# Redefining the ancestral origins of the interleukin-1 superfamily

Jack Rivers-Auty[1], Michael J.D. Daniels [1], Isaac Colliver[1], David L. Robertson [2,3] & David Brough [1]

The interleukin-1 (IL-1) receptor and ligand families are components of the immune system. Knowledge of their evolutionary history is essential to understand their function. Using chromosomal anatomy and sequence similarity, we show that IL-1 receptor family members are related and nine members are likely formed from duplication and modification of a proto-IL-1R1 receptor. The IL-1 ligands have a different evolutionary history. The first proto-IL-1β gene coincided with proto-IL-1R1 and duplication events resulted in the majority of IL-1 ligand family members. However, large evolutionary distances are observed for IL-1α, IL-18 and IL-33 proteins. Further analysis show that IL-33 and IL-18 have poor sequence similarity and no chromosomal evidence of common ancestry with the IL-1β cluster and therefore should not be included in the IL-1 ligand ancestral family. IL-1α formed from the duplication of IL-1β, and moonlighting functions of pro-IL-1α acted as divergent selection pressures for the observed sequence dissimilarity.

[1] Division of Neuroscience and Experimental Psychology, School of Biological Sciences, Faculty of Biology, Medicine and Health, Manchester Academic Health Science Centre, University of Manchester, AV Hill Building, Oxford Road, Manchester M13 9PT, UK. [2] Division of Evolution and Genomic Sciences, School of Biological Sciences, Faculty of Biology, Medicine and Health, University of Manchester, Michael Smith Building, Oxford Road, Manchester M13 9PT, UK. [3] MRC University of Glasgow Centre for Virus Research, Sir Michael Stoker Building, 464 Bearsden Road, Glasgow G61 1QH Scotland, UK. Correspondence and requests for materials should be addressed to D.B. (email: david.brough@manchester.ac.uk)

nflammation is an important host-response of the innate immune system to infection and injury. It is coordinated by soluble signalling and adhesion molecules that regulate cellular processes via cell surface and cytosolic receptors to neutralise infection or repair tissue injury[1,2]. The interleukin-1 (IL-1) family of cytokines and corresponding receptors constitute one of the main signalling components of inflammation[1,2]. It is commonly reported that there are 11 members of the IL-1 ligand family[1–3] and 10 members of the IL-1 receptor family[4,5]. The concepts of protein and gene families originally arose from evolutionary analysis where individuals are grouped into families based on shared common ancestry[6–8]. This definition has extended to

**Fig. 1** Evolutionary history of the IL-1 receptor family. **a** Chromosomal gene location as evidence of ancestral relationship of IL-1 family members. The highly conserved nature of surrounding genes and sequence homology suggests two or one separate evolutionary families: *IL-1R1, IL-1R2, IL-18R1, IL-1RL2, IL-1RL2* and *IL-18RAP* all form a cluster on the same chromosome, indicating gene duplication events. *IL-1RAPL2* also has the *MAP4K4* gene nearby indicating a duplication and translocation of that part of the chromosome. The *ARX* gene indicates that *IL-1RAPL1* likely formed from a duplication of *IL-1RAPL2*. *IL-1RAP* shares no nearby genes but demonstrates high sequence identity with the *IL-18RAP* gene inferring homology. This suggests there is clear evidence that all the IL-1R superfamily genes have a common ancestor except for the *SIGIRR* gene, which has relatively low sequence identity and no chromosomal anatomy evidence to support shared ancestry. **b** Composite evolutionary history of the IL-1 family of cytokines constructed by overlaying the evidence from chromosomal location and clade IL-1 family gene profile on to the maximum likelihood tree from Supplementary Data 1. Dotted lines represent less established evidence of shared ancestry. The percentage of trees in which the associated group clustered together is shown next to the branches from 1000 bootstrap replications. Branches, which occur in <50% of trees were collapsed. The tree is drawn to scale, with branch lengths measured in the number of amino acid replacements per site. The analysis involved 231 amino acid sequences with an alignment length of 64 positions in the final data set. All positions containing gaps and missing data were eliminated. Scale bar is 0.5 replacements per site

allow families to include functionally similar proteins that do not share common ancestry[1,3,6–8]. The distinction between the ancestral and functional definition is often delineated by the following terms: family and superfamily, respectively, which we use here[9]. The ambiguity of the term family has resulted in an unclear picture of IL-1 evolution with many inferring common ancestry without substantial evidence[10]. Indeed, when the "IL-1F" nomenclature of the IL-1 ligand family was defined, it was stated that "The IL-1 portion of the name is maintained to indicate the evolutionary relationship to the traditional types of IL-1"[10]. Understanding the evolutionary history of ligands and their receptors can provide critical insights in terms of biological relevance and is a requisite for effective research frameworks. Structural interactions between IL-18 and IL-18Rα, and IL-33 and IL-1RL1, were modelled using the established structural relationship of IL-1β and IL-1R1 as a search model[11,12], an approach that assumes commonality in structural relationships due to shared common ancestry of the ligands and receptors. Therefore, an in-depth investigation into these evolutionary relationships is needed.

Six members of the IL-1 receptor family are present in a cluster on the same chromosome in most clades providing strong evidence that these members formed through a series of tandem gene duplications of a proto-IL-1R1 gene prior to the divergence of the vertebrate clade[13]. On separate chromosomes, the evolutionary ancestry of IL-1 receptor accessory protein-like 2 (IL-1RAPL2), IL-1RAPL1, IL-1RAP and single Ig IL-1-related receptor (SIGIRR), is less clear and therefore, a critical investigation of the ancestral origin of these proteins is required[14]. Similarly, nine IL-1 ligand family members occur in a single cluster on human chromosome two and likely formed through a series of gene duplications of the prototypical IL-1 family cytokine IL-1β[2,15–17]. In this way, IL-1β, IL-1α, IL-36α, IL-36β, IL-36γ, IL-36RA, IL-37, IL-38, and IL-1RA are accurately described as family members with shared common ancestry[17–23]. IL-18 and IL-33, however, are present on different chromosomes and have low sequence identity, indicating weak evidence for evolutionary relatedness, i.e., homology to IL-1β. Yet they have been included into the IL-1 family based largely on structural similarities, specifically, the presence of a 12 beta- sheet trefoil fold, as well as overlap in function and the receptors involved[8,24–28]. However, a thorough investigation into whether IL-18 or IL-33 likely share a common ancestor with the members of the IL-1β cluster has not been performed[25].

An additional quandary of IL-1 ligand family evolution is the retention of the IL-1β paralogue IL-1α. IL-1β and IL-1α both signal through the type 1 IL-1 receptor (IL-1R1)[2,29], and are both produced as precursors (pro-forms) in cells of the innate immune system in response to an inflammatory stimulus such as a pathogen-associated molecular pattern (PAMP, e.g., LPS), or a damage-associated molecular pattern (DAMP, e.g., HMGB1)[2].

Both IL-1β and IL-1α employ unconventional secretory routes, which involve a cleavage step[30,31]. Pro-IL-1β is cleaved directly by the protease caspase-1 in the cytosol following activation by multi-molecular inflammasome complexes[32]. The processing of pro-IL-1α is thought to be regulated by calpains and is indirectly regulated by inflammasome/caspase-1 complexes[33]. While these mechanisms of activation may appear substantially different, they share the common feature of the loss of membrane homeostasis, which results in both potassium efflux essential for inflammasome formation, and calcium influx, leading to calpain activation[34–37]. Because of this, IL-1β and IL-1α have largely been considered as similar inflammatory cytokines released following membrane permeabilisation and cell death[38,39].

While IL-1β is known to have a specialised role as an inflammatory cytokine, it has been suggested that IL-1α is a moonlighting protein, which may function as an extracellular cytokine, and within the cell[40–43]. Interestingly, these functions have been proposed to involve the uncleaved pro-form of IL-1α. Several suggested functions occur in the nucleus as it is known that the pro-domain of IL-1α contains a nuclear localisation sequence (NLS), whereas pro-IL-1β does not[44]. This results in markedly different subcellular distributions of pro-IL-1α and pro-IL-1β, with pro-IL-1α localising strongly to the nucleus[44,45]. The moonlighting functions of nuclear pro-IL-1α have been suggested to bind to histone acetyltransferase (HAT), regulate gene transcription[42], regulate mRNA splicing[41], and nuclear sequestration as a mechanism to regulate secretion[45,46]. Unique features of pro-IL-1α have also been proposed outside the nucleus, including HCLS1 (hematopoietic cell-specific Lyn substrate 1)-associated protein X-1 (HAX-1) binding, function at the IL-1R1 receptor and cytosolic IL-1R2 binding[45–53].

Phylogenetic analysis allows us to infer evolutionary relationships between proteins and, based on evolutionary understanding, to develop new hypotheses. Here we used a phylogenetic analysis to examine the IL-1 ligand and receptor families with a particular focus on evolutionary evidence for the inclusion or exclusion of SIGIRR, IL-33 and IL-18 from their respective ancestral families. We also investigate IL-1α, a gene duplicate of IL-1β with overlapping functions despite having relatively dissimilar sequences.

## Results

**Evolution of the IL-1 receptor family**. There is strong sequence and chromosomal anatomy evidence that IL-1R1, IL-1R2, IL-1RAP, IL-1RL1 (ST2), IL-18R1, IL-1RL2 and IL-18RAP are members of the same family formed from ancestral gene duplications of a common proto-IL-1R (Supplementary Data 1, Fig. 1b). The close proximity to the *MAP4K4* gene of this IL-1R1 sub-family and the presence of a duplicate *MAP4K4* gene beside the *IL-1RAPL2* gene in the reptile and cartilaginous fish clades suggest that IL-1RAPL2 formed from a duplication and

translocation of a member of the IL-1R1 sub-cluster (Fig. 1a). The proximity of the *ARX* gene to the *IL-1RAPL2* gene in the cartilaginous fish and the *IL-1RAPL1* gene in the bird and mammal clades suggests that *ILRAPL1* formed from a duplication and translocation of the *IL-1RAPL2* gene (Fig. 1a). There is no chromosomal anatomy evidence to support the inclusion of *IL-1RAP* gene as an IL-1 receptor ancestral family member (Fig. 1a). However, the sequence conservation strongly indicates that *IL-1RAP* likely formed from a duplication event of *IL-18RAP* (Supplementary Data 1, Fig. 1b). The *SIGIRR* gene has relatively low sequence identity to the IL-1R family members as seen by the large branch distance (Supplementary Data 1, Fig. 1b), and there is no chromosomal evidence for common ancestry (Fig. 1a). Its function as a negative regulator of the IL-1R1 and TLR4 would place it as part of the IL-1R/TLR superfamily. As all members of the IL-1R superfamily except IL-1RAPL1 are present in all vertebrates it is likely that that these genes diverged prior to separation of bony and cartilaginous fish ~420 million years ago (Supplementary Data 1, Fig. 1a, b)[54,55], while IL-1RAPL1 likely formed from a gene duplication of IL-1RAPL2's ancestor prior to the separation of bony fish and the Tetrapoda clade 365 million years ago[54,55].

The IL-1RL1 (ST2/IL-33 receptor) is found in birds, fish and reptiles (Supplementary Data 1, Fig. 1a, b), while the ligand, IL-33, is only found in mammals (Supplementary Data 2, Fig. 2a–d). This suggests that perhaps there are alternative ligands in the other clades, or that another member of the IL-1 super family acts through IL-1RL1 receptor. The ligand-binding domain is not conserved across clades, indicating that differences in ligand are probable[56]. Ligand-independent functions have been reported with fish IL-1RL1 inhibiting TLR activity[56], suggesting an interesting evolutionary history of IL-1RL1 and the IL-33 ligand where the receptor had a function without a ligand, resulting in an extracellular domain that was able to mutate without altering the constitutive activity of the receptor. In the pre-mammalian clade Synapsids, these mutations resulted in amino acid residue changes, leading to the binding of the IL-33 ligand and a new ligand receptor relationship was formed.

The large sequence divergence in IL-1R2 relative to the rest of the IL-1R1 family is likely due to the lack of the Toll/interleukin-1 receptor homology domain (TIR, the intracellular proportion of the receptors) (Supplementary Data 1, Fig. 1b). The strong chromosomal anatomy evidence supports common ancestry, indicating an ancestral duplication event of the IL-1R1, followed by a truncation (Fig. 1a). Conservation of the gene was conferred based on its function as a decoy receptor. We see no marked evolutionary distance between mammalian and non-mammalian IL-1R2, which might have been expected given the proposed interactions between IL-1R2 and pro-IL-1α[53].

**Evolution of the IL-1 ligand family**. Inclusion into the IL-1 ligand family has previously largely been defined by similarities in structural homology, receptor binding and immunomodulatory function. An in-depth investigation into whether common evolutionary ancestry is responsible for this homology of structure and function has not been previously performed. Thus, it is unknown whether our current understanding of the IL-1 family accurately describes the relationship between IL-1 family members. To address this, we constructed a comprehensive phylogenetic tree using 155 sequences from across the animal kingdom and IL-1 superfamily ligand members. From this, we found IL-1β present in all vertebrate species and gene loci evidence, particularly the proximity to the *CKAP2L*, *SLC20A1*, *PSD4*, and *OGDH* genes, suggested that these IL-1β genes are orthologues retaining similar functions and diverging from a common proto-*IL-1β* gene

present in a common ancestor (Fig. 2a)[57]. This places the first appearance of IL-1β ~420 million years ago around the emergence of the vertebrate subphylum (Fig. 2b)[54,55]. We identified four distinct clusters separated by large evolutionary distances, including a primary group containing IL-1β, IL-1RA, the IL-36 subgroup, IL-38 and IL-37, as well as three distinctly separate groups of IL-18, IL-33 and IL-1α. IL-1α and IL-33 subgroups are exclusively mammalian placing their appearance after the divergence of the Sauropsida (ancestral lineage of reptiles, turtles and birds) and Synapsid (ancestral lineage of mammals) clades ~320 million years ago and prior to the divergence of mammalian species ~160 million years ago (Fig. 2b)[54,58], while the IL-18 subgroup is similar to IL-1β in that it is expressed in all vertebrates and likely appeared around 420 million years ago[54].

Evidence from chromosomal anatomy across clades and sequence similarity was used to examine the relationship of the four IL-1 family clusters to create a more realistic IL-1 ligand superfamily tree that better reflects the uncertain evolutionary history of protein/gene families, despite the inherently weak phylogenetic signal (Fig. 2d). From this, we concluded that while IL-1α very likely evolved from a common ancestor to IL-1β, and therefore, does belong in the IL-1 ancestral family, it appears that IL-18 and IL-33 do not. These analyses suggest that the IL-1β family cluster on human chromosome 2, IL-18 and IL-33 evolved in unrelated distinct evolutionary events. BLAST searches using human IL-18 and IL-33 gene sequences (NCBI Reference Sequence: NC_000009.12) against non-mammalian sequences reveal no sequences with even remote evidence of homology leaving the origins IL-18 and IL-33 unknown. Previously, the similarity in protein structure was considered sufficient evidence for common ancestry/homology to be inferred, yet, the defining feature of the IL-1 ligand superfamily, the beta trefoil fold, has occurred in a number of seemingly unrelated proteins across kingdoms. Indeed, previous analyses by Murzin et al. revealed that a wide range of sequences are capable of forming these structures, supporting the occurrence of independent evolutionary events[24,59]. Furthermore, the likelihood of such convergent evolution is inversely proportional to the complexity and specificity of the protein structure; as the trefoil fold is relatively simple and flexible the probability of independent evolution is reasonable[59]. The Separate evolution of IL-18 and IL-33 was further supported by additional sequence analyses of the functionally unrelated, but structurally related, proteins of the fibroblast growth factor (FGF) family, which contain a beta trefoil fold (Supplementary Data 3). From this it was found that IL-18 and IL-33 are equally unrelated to both the IL-1 ancestral family and the FGF family. The constructed tree of 344 sequences indicated equal sequence dissimilarity between IL-18 and IL-33 to the FGF and IL-1 ancestral families and shows great instability as expected from a constructed tree of unrelated proteins (Supplementary Data 3). Furthermore, the stability and confidence of the maximum likelihood trees was greatly increased when reanalysed as three separate trees of the IL-1β family cluster on human chromosome 2, IL-18 and IL-33 (Supplementary Data 6–8). Thus, we argue that the use of the term "IL-1 family" is misleading in the case of IL-18 and IL-33 and that the term superfamily better describes our current understanding of the evolutionary history of the IL-1 superfamily members (Fig. 2c, d).

Chromosome gene anatomy evidence and intron/exon structure homology strongly suggests that IL-1α likely arose as a result of an ancestral gene duplication of IL-1β between 320 and 160 million years ago (Supplementary Data 2, Fig. 2a–d)[54,57,60,61]. However, the sequence of IL-1α appears as to be as almost as unrelated to IL-1β as IL-18 and IL-33 (Supplementary Data 2, Fig. 2d). We hypothesise this dissimilarity between the sequences of IL-1α and IL-1β is due to sub-functionalization of IL-1α, i.e.,

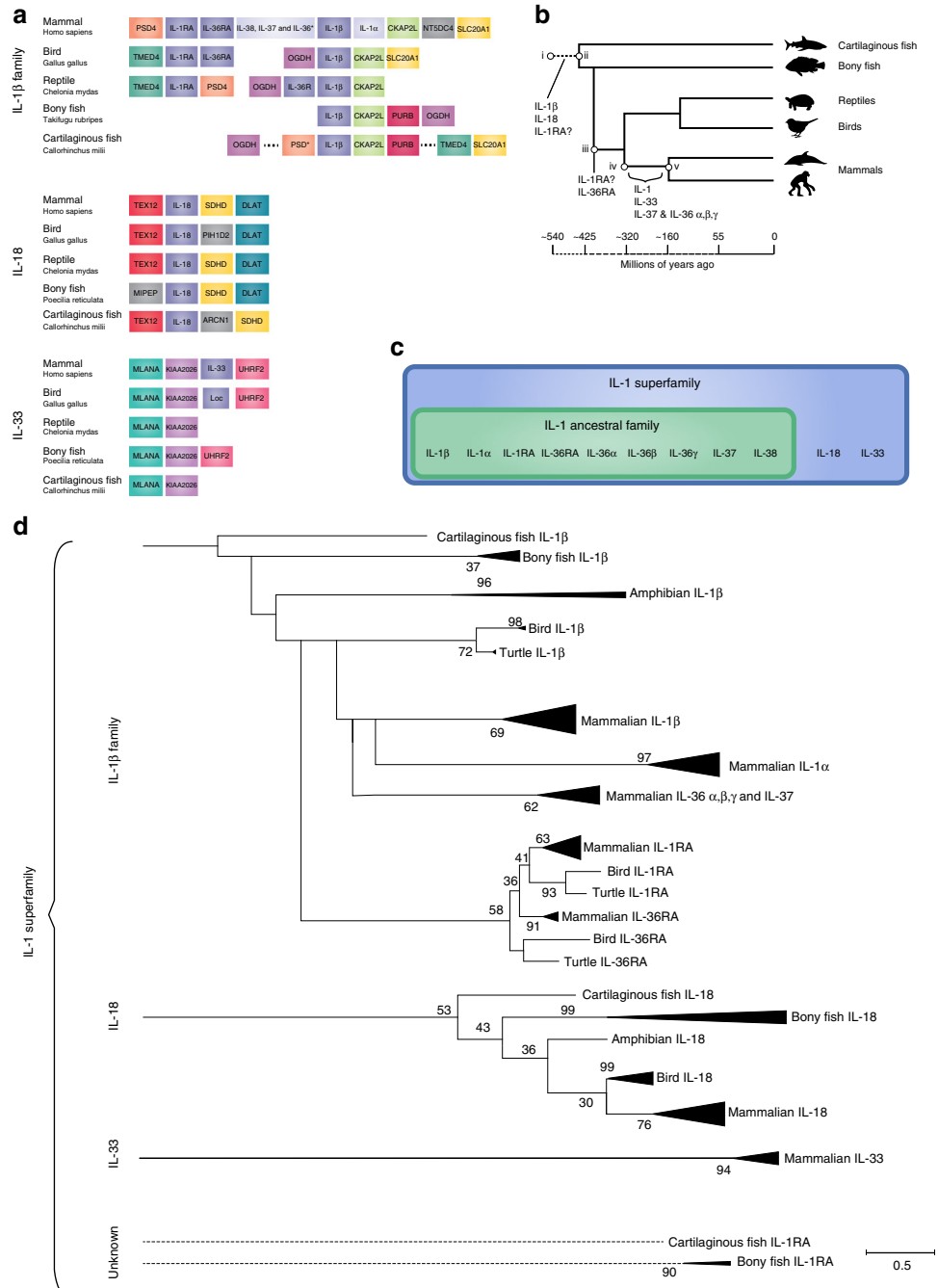

**Fig. 2** Evolutionary history of the IL-1 ligand family. **a** Chromosomal gene location as evidence of ancestral relationship of IL-1 family members. The highly conserved nature of surrounding genes suggests three separate evolutionary families, as follows: IL-1β, IL-18, and IL-33. **b** A simplified evolutionary tree with time scale of cartilaginous and bony fish, birds, reptiles, mammals and the approximate time line of the evolutionary occurrence of IL-1 family members[54,55]. IL-1β and IL-18 are expressed exclusively in all vertebrate species, including cartilaginous fish, suggesting that they evolved prior to the divergence of bony and cartilaginous fish ~425 million years ago (ii); IL-1α, IL-33 and IL-36 α, β & γ are expressed exclusively in mammals therefore likely formed in the common ancestor of all mammals (Synapsid lineage) (iv). This event must have occurred after divergence of the Synapsid lineage (iv) 320 million years ago but prior to the divergence of mammals 160 million years ago (v). **c** An ancestral and superfamily scheme of the IL-1 ligands. **d** Composite evolutionary history of the IL-1 family of cytokines constructed by overlaying the evidence from chromosomal location and clade IL-1 family gene profile on to the maximum likelihood tree from Supplementary Data 2. The percentage of trees in which the associated group clustered together is shown next to the branches from 1000 bootstrap replications. Primary branches that occur in <50% of trees were collapsed. The tree is drawn to scale, with branch lengths measured in the number of amino acid replacements per site. The analysis involved 155 amino acid sequences with a final alignment length of 64 positions. All positions containing gaps and missing data were eliminated. Scale bar is 0.5 replacements per site

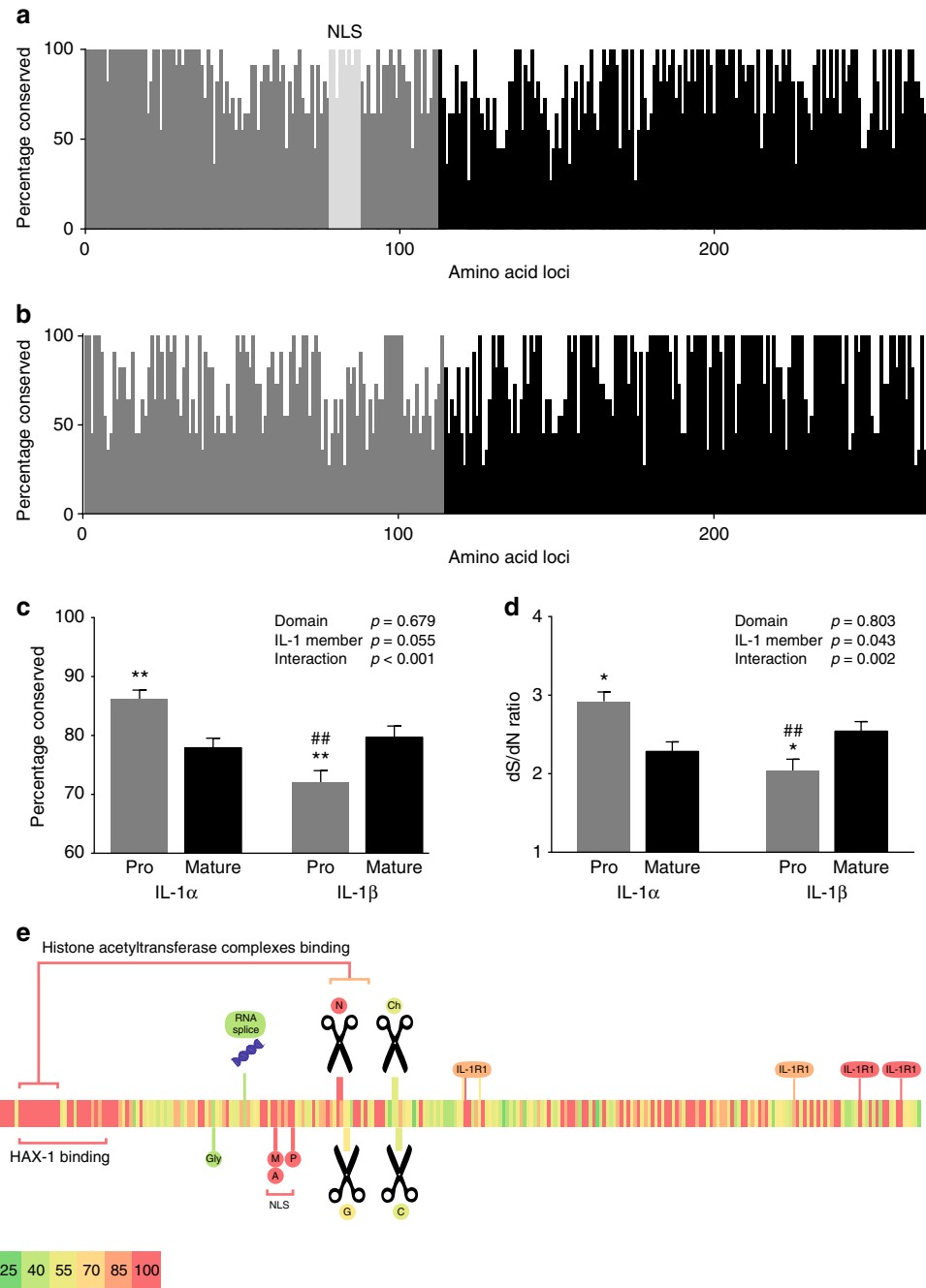

**Fig. 3** Conservation of IL-1α and IL-1β sequences in mammals. The percentage of conserved a.a. residues of the pro (grey) and mature (black) segments of IL-1α (**a**) and IL-1β (**b**). **c** The average conservation from **a**, **b** showing that the amino acid sequence of the pro-domain of IL-1α is highly conserved, more so than the mature domain of either IL-1α or IL-1β. **d** The ratio of synonymous (dS) to non-synonymous (dN) substitutions per potential substitution site demonstrates strong positive selection for the conserved pro-domain of IL-1α. Analyses were conducted using the Nei-Gojobori method[80]. **e** A heat map of the conservation of IL-1α sequence across mammalian species depicting the correlation with putative functions of the IL-1α protein (green 25% conserved to red 100% conserved across mammalian species). The NLS sequence, histone acetyltransferase complex-binding region, HAX-1-binding region, neutrophil elastase cleavage site (N) and NLS post-translation modification sites were very highly conserved (P, phosphorylation; A, acetylation; M, myristoylation), while the glycosylation (gly) site, RNA splicing domain and cleavage sites for chymase (Ch), caspase-1 (C), and granzyme B (G) were poorly conserved. **c**, **d** bars are means ± SEM, $*p < 0.05$ and $**p < 0.01$ pro-domain vs mature domain within IL-1 member; $\#p < 0.05$ and $\#\#p < 0.01$ IL-1β vs. IL-1α within domain, omnibus effects evaluated by linear modelling followed Sidak correct post-hoc tests

divergent evolutionary pressure associated with the distinct functions. Therefore, we performed further conservation analysis at the protein domain level and found that there are at least two potential selective forces driving the divergence of IL-1α, both involving neo-functionalization of the pro-domain (Fig. 3).

**Homology of the pro-domains of IL-1α and IL-1β**. To investigate the divergence of IL-1α from IL-1β, amino acid sequence similarity was compared across the sequence of each protein in 11 mammalian species (Fig. 3). From this, we found that while the mature domains of IL-1α and IL-1β are similarly conserved, we

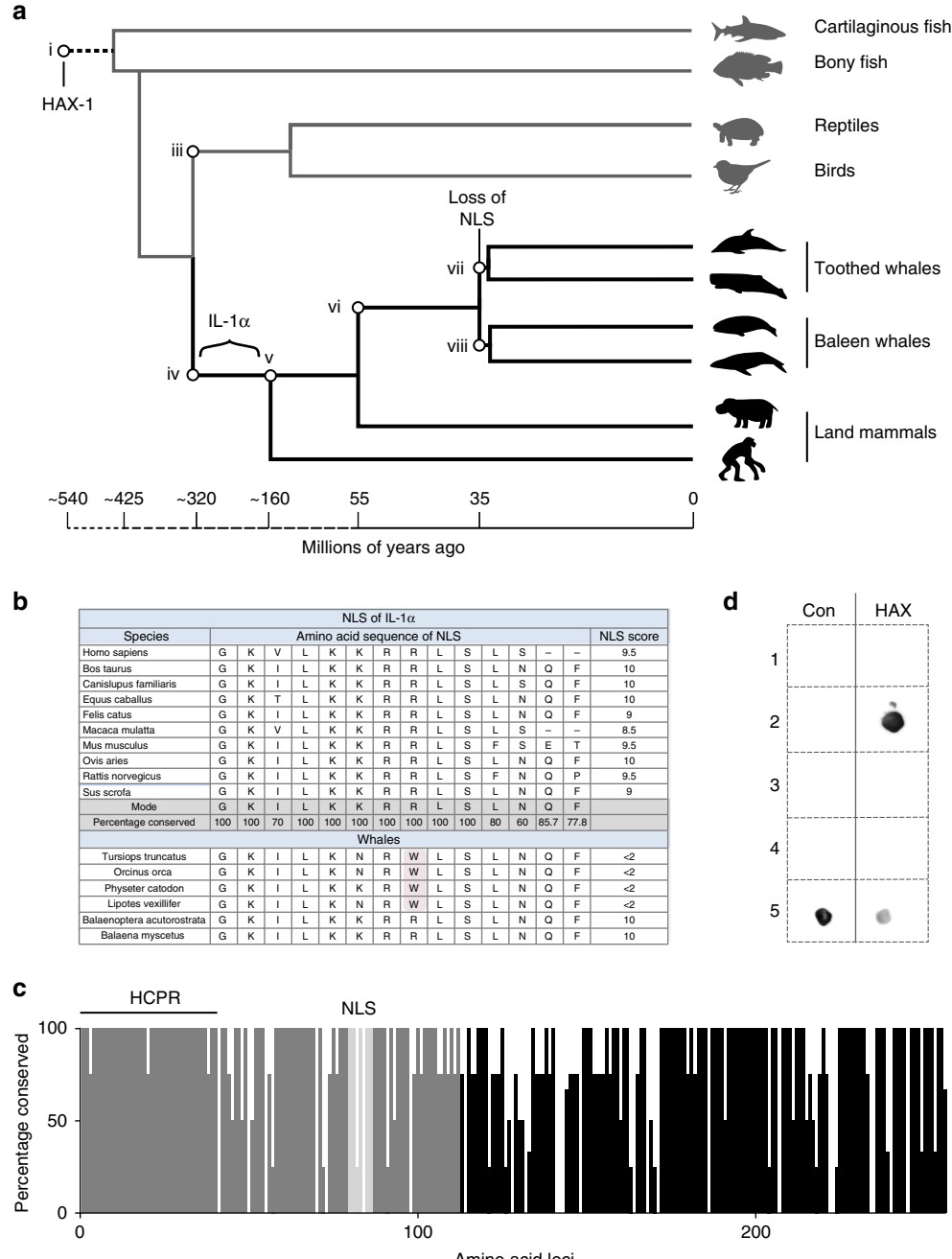

**Fig. 4** The evolutionary importance of HAX-1 to IL-1α. **a** A model evolutionary tree with time scale of cartilaginous and body fish, birds, reptiles, mammals and baleen and toothed whales[54,55]. HAX-1 is expressed in all vertebrate species as well as hemichordates, suggesting that it emerged during the Cambrian explosion (i); IL-1β evolved approximately 425 million years ago (ii); IL-1α formed via a gene duplication event of IL-1β in the common ancestor of all mammals (Synapsid lineage) (iv). The NLS of IL-1α is present in all land mammals and baleen whales, suggesting it was present in the common ancestor of all mammals (v) and the common ancestor of all whales (vi); all toothed whales lack a functional NLS, suggesting that the common ancestor of the toothed whale (vii) had a loss of function mutation in the NLS. **b** The amino acid sequences of the IL-1α NLS, including the percentage conservation and the calculated predicted NLS activity (cNLS) scores[63]; all found in the pro-region in amino acid positions around 70–85. Note the loss of a functional NLS in the toothed whale species due in most part to the R to W replacement (highlighted). **c** The percentage of conserved amino acid sequences of the pro (grey) and mature (black) segments of IL-1α comparing 4 toothed whale species with the modal land mammal sequence. While the mature segment contains substantial variation due to the build-up of neutral mutations over the 35 million years of evolutionary separation, the pro-IL-1α domain is highly conserved despite the loss of NLS function. **d** Dot blot with bovine serum albumin (BSA) (1), pro-IL-1α (2), mature IL-1α (3), mature IL-1β (4) and HAX-1 (5) bound to the membrane (0.02 μg), then incubated with HAX-1 (5 μg ml$^{-1}$) (HAX) or control buffer (Con) to investigate HAX-1 binding. HAX-1 was probed and visualised, and found to exclusively bind to pro-IL-1α, confirming that this binding is specific and substantial to the pro-domain of pro-IL-1α

see completely different trends in the pro domains of each protein (Fig. 3a, b). The conservation of the pro-domain of IL-1β is lower than the mature domain as well as the ratio of synonymous to non-synonymous nucleotide mutations, which is consistent with a domain that is of lower functional importance and supporting that the sole role of the pro-domain of IL-1β is to prevent binding to IL-1R1 (Fig. 3b)[62]. This is starkly contrasted by IL-1α whose pro-domain is more highly conserved and has a higher synonymous/non-synonymous ratio than the mature domain (Fig. 3c, d), suggesting that the pro domain has a specific function, and this caused divergence from IL-1β and constrained the sequence variability of IL-1α across species. In particular, within the pro domain there are two subregions that are highly conserved: the 1–40 amino acids highly conserved pro-domain region (HCPR) and the NLS (Fig. 3e).

**The role of the NLS in neo-functionalisation of IL-1α**. Using the nuclear localisation ten-point scoring system developed by Korsugi et al.[63] based on known amino acid sequences critical to the classical importin αβ pathway to compute predicted NLS activity (cNLS), we compared a diverse selection of mammalian IL-1α pro-domain sequences. Surprisingly, the toothed whale clade (Odontoceti) lacked a functional NLS with cNLS scores less than two[63] (Fig. 4b). This is largely due to the replacement of the key positively charged amino acid arginine to the uncharged hydrophobic amino acid tryptophan (KKRR to KKWR) (Fig. 4b). This provided a unique opportunity to use conservation and evolutionary evidence to test the hypothesis that the key moonlighting function driving IL-1α divergence resides in the nucleus such as HAT binding. This hypothesis would predict that a number of amino acid residue replacements would increase in the pro-domain in the toothed whale clade as without an NLS the nuclear moonlighting function would now be redundant and so purifying selection will be relaxed. To evaluate this, we compared the IL-1α sequence of the four toothed whale species, which have published genomes with the modal sequence of land mammals (Fig. 4c). From this analysis, we find that the pro-domain remains highly conserved despite the non-functional NLS (Fig. 4c). The functional mature domain of IL-1α has a substantial 'build-up' of amino acid replacements, suggesting that the evolutionary time since the toothed whale diverged was similarly sufficient for changes to accumulate (Supplementary Data 5), yet this had not occurred in the HCPR of IL-1α presumably due to the action of purifying selection (Fig. 4a, c). This does not support the hypothesis that there is a nuclear moonlighting function driving IL-1α divergence from IL-1β, indicating that nuclear functions such as HAT binding are not likely to be the major force driving the divergence of IL-1α.

HCLS1 (hematopoietic cell-specific Lyn substrate 1)-associated protein X-1 (HAX-1) most likely appeared during the Cambrian explosion ~540 million years ago (Supplementary Data 4) as it is present in all vertebrate species and in distantly related animals such as arthropods (Fig. 4a(i))[54]. It is a cytosolic protein that has a number of structurally unrelated binding partners, including HS1 (hematopoietic lineage cell-specific protein-1), cortactin, PKD2 (polycystic kidney disease-2/polycystin-2), EBNA-LP (Epstein–Barr nuclear antigen leader protein) and Bcl-2. The exact function of HAX-1 binding is currently unknown. However, it has been linked to the shuttling of proteins to the nucleus or mitochondria, as well as masking active domains on the protein (for review[51]). After comparing the conservation of IL-1α with the putative moonlighting functions of pro-IL-1α, and taking into consideration, the conservation of each subdomain across the toothed whale clade which lack an NLS, our evolutionary evidence suggests that HAX-1 binding is the major directional

selective force pushing the divergence of IL-1α from IL-1β (Figs. 3e and 4c). Using dot blots a biologically specific interaction between pro-IL-1α and HAX-1 was confirmed (Fig. 4d). HAX-1 interacted with pro-IL-1α, and no interaction with mature IL-1α or IL-1β was observed (Fig. 4d). This confirmed the specificity of the interaction and supports the evolutionary evidence, suggesting an interaction with HAX-1 drove divergence of pro-IL-1α.

**Discussion**
The IL-1 ligand and receptor superfamilies are involved in a number of functions, primarily immune modulation[1,2]. Evolutionary analysis was carried out to investigate the origins and key functions of these cytokines and their receptors. From this, we found that the IL-1R family is a conserved group of receptors that have existed for at least 420 million years[54]. Except for the SIGIRR receptor, all receptors exhibit strong evidence to have formed through ancestral gene duplication events. The SIGIRR receptor is the only described family member that does not have evidence of IL-1R ancestral lineage, suggesting that SIGIRR should be considered a superfamily IL-1R member and not an ancestral family member. The IL-33 receptor (IL-1RL1) is present in the bony fish, reptilian and avian clades as well as mammals, despite non-mammalian clades lacking the IL-33 ligand. This suggests the presence of other ligands for IL-1RL1, or functionality as an orphan receptor. Work by Reble et al.[56] supports the latter hypothesis finding that rainbow trout IL-1RL1 acts constitutively as a IL-1RL1 and TLR4 pathway inhibitor by sequestering the secondary signalling molecule MYD88[56]. We hypothesise that, without a ligand, the extracellular portion of IL-1RL1 lacked stabilisation selection pressure and this caused accelerated mutagenesis of this region allowing binding of a novel signalling molecule. This process where the receptor "found" the ligand, as opposed to the common ligand and receptor co-evolution from their respective family members[59], may explain the lack of shared common ancestry of the IL-33 and IL-1β ligands.

The IL-1 ligand superfamily has four distinct subgroups, including the primary cluster whose central member is IL-1β as well as the IL-33, IL-18, and IL-1α subgroups. Using chromosome anatomy and sequence analyses, we conclude that IL-33 and IL-18 do not share a common ancestor to the other IL-1 family members and therefore should not be included in the IL-1 ancestral family, but based on structural fold and receptor homology should be considered members of the IL-1 superfamily. The IL-1α cluster appears to have emerged as a consequence of a duplication event around the emergence of the Synapsid proto-mammalian clade. Although IL-18, IL-33, and IL-1α appeared approximately equally unrelated to IL-1β at the sequence level (Supplementary Data 2), chromosome gene anatomy evidence and intron/exon structure homology strongly suggested that IL-1α likely occurred as a gene duplication of IL-1β between 320 and 160 million years ago, while IL-18 and IL-33 occurred in separate evolutionary events (Fig. 2b)[57,60,61]. This raised the question, why do such functionally and structurally similar proteins arising from gene duplication have such dissimilar sequences? This dissimilarity is particularly apparent when we compare the close evolutionary distance and sequence similarity of bird, turtle, and mammalian IL-1RA and IL-36RA, which are evolutionary more ancient (Supplementary Data 2). Our conclusion was that IL-1RA and IL-36RA evolved under strong purifying selection forces due to specificity of function, while IL-1α was "pulled" by diversifying selection to be very different from IL-1β at the sequence level despite the apparent similarity in function and recent shared ancestral origins. This presence of a diversifying/disruptive

selective force is supported by evolutionary evidence, which demonstrates that duplicate gene divergence is typically very slow, especially if the genes remain in close proximity on the chromosome, particularly if sub-functionalization is evident; illustrating that the dissimilarity of IL-1β and IL-1α is shaped by a selection pressure associated with functional divergence[64].

By analysing the homology of IL-1β and IL-1α sequences across the domains of each molecule, we discovered that the pro-domain of IL-1α was highly conserved, more so than the mature domain of either molecule, while the pro-domain of IL-1β was least conserved. This suggested that the pro-domain of IL-1α has important moonlighting functions, which drove divergence and extended the evolutionary distance from IL-1β. We define a particularly conserved region of the N-terminal domain of IL-1α as the HCPR, and used sequence conservation evidence to investigate its putative roles.

There are several putative hypotheses of pro-IL-1α function and activity that may explain the HCPR and the active divergence from IL-1β. Early work has suggested that pro-IL-1α may be presented on membranes via glycosylation dependent mechanisms, potentially including lectin or heparin sulfate interactions[65,66]. However, lectin associations normally require glycosylation and neither pro-IL-1α nor pro-IL-1β contain signal peptides and are therefore unlikely to be exposed to the relevant glycosylation machinery. Furthermore, proteomic analysis only identifies two likely points of glycosylation in pro-IL-1α in the human sequence, and these are outside the HCPR at the amino acid loci of 102 and 141 in the human protein, and the amino acid composition of these sites are not highly conserved across mammals (Fig. 3e)[67]. Heparin sulfate is a non-conventional mechanism of lectin-like membrane association. However, heparin sulfate association requires basic amino acid clusters, and proteomic analysis shows that the HCPR is the most acidic region of the pro-IL-1α protein with a pI of 4.49, compared to the mature IL-1α domain which has a pI of 5.57[68,69]. Therefore, while lectin-like extracellular membrane binding may have niche function in some mammalian species, it seems unlikely that this binding function is responsible for driving the divergence of IL-1α from IL-1β, and IL-1α retention.

Pro-IL-1α has been shown to be cleaved by a number of enzymes to generate products that are more active at IL-1R1 including calpains, gramzyme B, chymase and neutrophil elastase[70]. Comparing conservation of these cleavage sites, we see relatively low conservation of the calpain cleavage site, however, this is largely due to replacement of similar amino acids such as the amino acids serine and phenylalanine are replaced by the similar asparagine and tyrosine in some species, respectively (Fig. 3e)[71]. These changes do not affect calpain activity as the murine pro-IL-1α sequence contains these replacements and is cleaved by calpain into the mature form[72]. Similarly, granzyme B, neutrophil elastase and chymase cleavage sites are highly conserved and any variation in amino acid sequence is largely between functionally similar amino acids (Fig. 3e)[71]. Cleavage by these enzymes has been shown to produce an IL-1α product that is highly active at IL-1R1[70]. This suggests that pro-IL-1α could be released during tissue damage where it acts weakly as a DAMP, however, if there is sufficient damage or infection for recruitment of immune cells, which produce granzyme B (cytotoxic B-cells and T-cells), neutrophil elastase (neutrophils) and chymase (basophils and mast cells), the proinflammatory IL-1α signal will be amplified through pro-IL-1α cleavage into more active forms[70,71]. However, these cleavage sites represent a small number of amino acids that are variably conserved and are not present in the HCPR of IL-1α (Fig. 3e). Therefore, while these cleavage sites illustrate functional differences between IL-1α and

IL-1β, it is not likely that they are sufficient for the sequence divergence of IL-1α from IL-1β and IL-1α retention.

A further defining feature of IL-1α and IL-1β is their affinity for IL-1R2. While mature IL-1β has an affinity for IL-1R2 orders of magnitude greater than IL-1α, one study reports the pro-IL-1α has biologically relevant affinity for IL-1R2[53]. This binding can occur in the cytosol of a cell and prevent cleavage into the more active mature form. However, our receptor analyses (Fig. 1) suggests that this binding is not driving the divergence of IL-1α as this would involve a co-evolution of both the IL-1R2 and IL-1α. This should cause a greater drift in the IL-1R2 sequence in mammals from the non-mammalian clades[59]. However, there is no evidence of the drift, suggesting this binding, while probably biologically relevant in a subset of mammalian species, has not been a major driving force in IL-1α divergence from IL-1β. This conclusion is further supported by Kawaguchi et al. 2006 who demonstrated that the binding of pro-IL-1α to IL-1R2 is dependent on the mature domain of IL-1α, suggesting that the selective pressure constraining the sequence of the pro domain of IL-1α is not IL-1R2 binding[49].

There is accumulating evidence implicating specific functions of IL-1α in the nucleus[41,45–47,50,73]. A study by Pollock et al.[41] reported that pro-IL-1α can induce apoptosis via a Bcl2 (B-cell lymphoma/leukemia-2)-dependent mechanism by modulating RNA processing apparatus. However, this region is not highly conserved (Fig. 3e), making it unlikely that this function was a major contributor to IL-1α divergence. However, the existence of one or more moonlighting functions in the nucleus is to some extent supported by our conservation analyses. First, we see that the NLS is highly conserved across distantly related mammalian species along with the post-translational modification sites in and around the NLS, which may have important roles in modulating its function (Fig. 3e)[74]. Second, there is high conservation of the regions that bind the histone acetyltransferase (HAT) complexes and these domains overlap with the HCPR (Fig. 3e)[47]. Research by Cohen et al.[48] reported pro-IL-1α induces expression of a number of inflammatory cytokines, including IL-6 and IL-1α itself, through a mechanism that is NLS and HAT complex-binding dependent[48]. The role of HAT complex binding in the divergence of IL-1α from IL-1β is also supported by modes of evolution observed in other moonlighting proteins[75]. This postulates that the ultimate fate of duplicate genes where one gains a moonlighting function is complete specialisation of each gene, hence, the moonlighting function becomes the primary function of the duplicated gene unless the moonlighting function of the gene duplicate is synergistic with the ancestral function[40,46,75]. The latter does appear to be true for IL-1α with both its IL-1R1 activity and HAT complex binding both inducing a proinflammatory response[40,46,75]. Furthermore, an evolutionary driver of moonlighting function is exposure to a new cellular environment[40,75]. For IL-1α, this may have been the emergence of the NLS in the proto-mammal, exposing IL-1α to a new cellular environment at considerably higher levels than prior to the development of the NLS[40,75]. This led IL-1α to develop novel moonlighting functions in a different cell compartment[40,75]. Therefore, the sequence analyses presented here, as well as previously published experimental evidence and evolution theory support the hypothesis that IL-1α diverged from IL-1β and was retained as a moonlighting protein, which directly modulates gene function as well as maintaining its ancestral function of IL-1R1 activity. However, the absence of a NLS in IL-1α from the toothed whale argues against a nuclear function driving the divergence of IL-1α. For this reason, we favour HAX-1 binding as the divergent pressure on IL-1α function and sub-functionalization. Yin et al.[52] demonstrated that pro-IL-1α and the pro fragment alone bind to HAX-1 and Kawaguchi et al.

expanded on this work establishing that HAX-1 binding to pro-IL-1α is important for intracellular proinflammatory effects of pro-IL-1α in fibroblast cells, demonstrating siRNA knock down of HAX-1 resulted in blunted IL-6 release[49,52]. We confirmed the physical and selective interaction between pro-IL-1α and HAX-1 here (Fig. 4d). Beyond these studies, the function of HAX-1 binding of pro-IL-1α, particularly in innate immune cells, has not been studied. The evolutionary evidence presented here suggests that our efforts to understand IL-1α biology should be refocused to this research area, as a substantial and important function of HAX-1 binding is likely. If HAX-1 binding is the key moon-lighting neofunction of IL-1α, we can hypothesise that IL-1α appeared via a gene duplication event of IL-1β after Synsapsid divergence 320 million years ago (Fig. 4a). Then, prior to mammalian divergence, pro-IL-1α accumulated mutations that included greater binding affinity to HAX-1, providing a neofunction of IL-1α and a selective advantage. This HAX-1 relationship drove the functional divergence of IL-1α from IL-1β, while the action of purifying selection prevented the accumulation of amino acid changes in the HAX-1 binding regions during the 160 million years of mammalian speciation (Fig. 4a). Evolutionary theory on moonlighting functions, suggests that it is likely that the function of the HAX-1/IL-1α binding is synergistic with inflammatory ancestral IL-1R1 function of IL-1α. However, the exact function of this important relationship has not been fully elucidated and therefore warrants future research.

In conclusion, our in-depth analyses into the evolution of the IL-1 ligand and receptor superfamily members has both improved our understanding of the ancestry of these genes and in this way redefined IL-1 ligand and receptor families to more accurately describe their evolutionary history, as well as providing key evidence into IL-1α neofunctionalization and therefore guide future research efforts.

## Methods

**Tree analysis**. Sequences were retrieved using BLASTN and BLASTP searches (http://www.ncbi.nlm.nih.gov/) with default parameters using established IL-1 family sequences. Supplementary Data 9 contains all gene names, clades and ids. Evolutionary history was inferred by using the maximum likelihood method based on the JTT matrix-based model[76]. The tree with the highest log likelihood is shown. Initial tree(s) for the heuristic search were obtained by applying the neighbor–joining method to a matrix of pairwise distances estimated using a JTT model. The coding data were translated assuming a standard genetic code table. All positions containing gaps and missing data were eliminated from alignments. For Supplementary Datas 1 and 2, a total of 66 and 64 positions were used, respectively, with a total of 231 and 155 amino acid sequences, respectively. Trees were reconstructed using the Whelan and Goldman, and Dayhoff matrix-based models and only subtle changes in the log likelihood values were observed and no inference altering differences were observed in the tree structures[77,78]. Evolutionary analyses were conducted in MEGA7[79]. The phylogenetic inference of these trees was intrinsically unreliable due to the short length of the alignments. This was caused by the inclusion of non-family members in the alignment supporting the conclusions of this study. Chromosomal anatomy was then used to improve the confidence in evolutionary history inference and trees containing only family members were constructed and presented in the supplement. These had longer sequence alignment and greater stability.

**NLS and conservation analysis**. Monopartite and bipartite NLS were identified and scored using NLS mapper[63]. Homology inferred from comparison of amino acid sequences of the pro, mature and NLS domains were calculated as percentage of species with the modal amino acids at each aligned site. All positions where the mode was a gap in the alignment were eliminated from the analysis.

**Synonymous/non-synonymous statistics**. The ratio of synonymous (dS) substitution per site to non-synonymous substitutions (dN) were conducted using the Nei-Gojobori method[80]. This method computes the numbers of synonymous and nonsynonymous substitutions and the numbers of potentially synonymous and potentially nonsynonymous sites. The count of the number of synonymous differences is normalised using the possible number of synonymous sites. A similar computation can be made for nonsynonymous differences. The Jukes–Cantor correction (dS or dN) computed were corrected to account for multiple substitutions at the same site. These analyses were conducted in MEGA7[79].

**Physical pro-IL-1α and HAX-1 interaction analyses**. HAX-1-binding was evaluated using the dot blot method. Strips of PVDF membrane were activated with methanol. Recombinant proteins (0.02 μg) were dotted onto the membrane in water and allowed to dry (HAX-1, Proteintech Ag27244; pro-IL-1α, Proteintech Ag10467; mature IL-1α, Abcam 200-LA; mature IL-1β, abcam 201-LA). The membranes were then washed three times with blocking buffer (5% BSA in PBS) and then left at room temperature (RT) for 2 h in blocking buffer. The membranes were exposed to 5 μg ml$^{-1}$ HAX-1 in PBS with 0.5% Triton x-100 or PBST alone for 3 h at RT, washed in blocking buffer, and then incubated in HAX-1 polyclonal antiserum (Abcam ab78939; 1:1000, 2 h at RT in 1% BSA PBST). The blots were washed, followed by goat anti-rabbit IgG conjugated to horseradish peroxidase (Dako P0448; 1:1000, 2 h in 5% milk PBST). Extensive washing was performed and bound antibody was then visualised by chemiluminescence.

**Statistical analyses**. Linear mixed modelling was used to evaluate the effect of independent factors on the dependent variable (nlme v1.19[81]). All factors and interactions were modelled as fixed effects. A within-subject design with random intercepts were used for all models and by-subject random slopes were applied where appropriate. The significance of inclusion of a dependent variable or interaction terms were evaluated using log-likelihood ratio. Holm-Sidak post-hocs were then performed for pair-wise comparisons using the least square means (LSmeans[82]). Homoskedasticity and normality were evaluated graphically using predicted vs residual and Q-Q plots, respectively. All analyses were performed using R (version 3.3.3).

**Data availability**. All data generated or analysed during this study are included in this published article and its supplementary information files.

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

## Acknowledgements

J.R-A. was funded by the BBSRC (BB/P01061X/1). MJDD is funded by a MRC DTP studentship (MR/K501311/1). D.L.R. was partially supported by the MRC (MC_UU_12014/12). D.B. was partially funded by the Wellcome Trust (083482/Z/07/Z).

## Author contributions

J.R-A and I.C. performed sequence and chromosomal analysis. J.R-A. and M.J.D.D. performed the biochemical experiments. D.L.R. and D.B. conceived, designed and supervised the study. J.R-A., D.L.R., D.B., analysed and interpreted data and wrote the manuscript.

## Additional information

**Competing interests:** The authors declare no competing interests.

