## [Peer Review File(PDF 786 kb) · Nature Communications]

Reviewer #1 (Remarks to the Author):

In the present study, Rivers-Auty et al. investigated the relationship between IL-1 family members using chromosomal anatomy and sequence homology. Based on evolution taxonomy, the authors show that IL-1 family members are grouped in 4 clusters, in part originated from duplication events of the prototype IL-1 β , which is present in all vertebrates. One cluster includes IL-1 β , IL-1RN, the IL-36 subgroup, IL-38 and IL-37, the others include IL-18, present in all vertebrates, and IL-33 and IL-1 α , which are found only in mammals.

Based on sequence homology, they propose that IL-18 and IL-33 should not be included in the IL-1 family.

They also show that the pro domain of IL-1 α is highly conserved, in contrast with the pro domain of IL-1 β , and they suggest it has a specific function which caused the divergence from IL-1 β and constrained the sequence variability of IL-1 α across species. Based on sequence homology among mammals, they propose that HAX-1 and histone acetyltransferase complex binding had a major role in the divergence of IL-1 α from IL-1 β .

This study is interesting and innovative since it raises novel hypothesis on the evolution of the IL-1 family and on selective forces driving the divergence of IL-1 α from IL-1 β that involve the neo-functionalization of IL-1 α pro-domain. However, it is highly speculative and no functional data supporting these hypotheses are reported.

Major comments:

-Concerning IL-18 and IL-33, the authors propose that structural similarity is not sufficient to include them in the IL-1 family, and indeed they show that sequence homology with the other members is low. Thus evolutionary evidence would suggest the exclusion of IL-33 and IL-18 from the IL-1 family. This analysis is important for better understanding the evolution and biology of these cytokines. However, the authors should take into consideration functional and biological homologies that justify or explain their inclusion in the IL-1 family (e.g. the homology among the receptors of IL-1 family, including IL-18R α and IL-18R β , or ST2 which couples with IL-1RAcP to signal; consider that IL-37 signals through IL-18R α , that all the members of the 4 clusters share a common mode of activation through cleavage of the pro domain and activate a shared signalling pathway).

-The authors propose that HAX-1 binding is the major directional selective force pushing the divergence of IL-1 α from IL-1 β , based on the high conservation of the regions which bind HAX-1 in IL-1 α pro domain. This hypothesis is novel and interesting. However the authors do not show functional data supporting it. The authors should comment on the relationship between HAX-1 binding and the described differences in function of IL-1 β and IL-1 α .

HAX-1 is introduced in the manuscript abruptly (line 292) not showing results, but discussing its potential relevance. Lines 301-303 should be presented first.

-They also propose that gene regulation through histone acetyltransferase complex binding may have been involved in pushing the divergence of IL-1 α from IL-1 β . It is not clear for the reader whether this hypothesis is considered solid to the authors.

Minor comments

- HCLS1 should be spelled out.
- Not all figure panels are mentioned in the text.

Reviewer #2 (Remarks to the Author):

Publication:

Redefining the interleukin 1 family from an evolutionary perspective
by Rivers-Auty, Colliver, Robertson and Brough

This publication aims to present a very important topic with possible future consequences in several areas of research but it needs to be substantially improved if a real redefinition of the interleukin 1 family (or superfamily?) shall take place.

Suggestion: major revision

Points to consider & improve:

Abstract, line 24: In content of an evolutionary publication it is rather vague to use the term "sequence homology". Either it is just only a sequence similarity or it is really an evolutionary homology. The chaotic usage of such terms was criticised previously many times by many reviewers in evaluation of papers aimed to describe evolutionary relationships. (Please consider this aspect also in the whole manuscript).

Abstract, line 26: The term "founding member" is really strange for a phylogenetic analysis. Better "basal clade" (if it really is one) or "monophyletic group" (if more clades are phylogenetically joined together leading to a common ancestor) shall be used - also throughout the whole manuscript.

Abstract, line 29: "no evidence of common ancestry" - brief details for this shall be given in the abstract

Introduction line 39: "There are 11 members of the IL-1 family"... so only eleven is really a very few in times where hundreds of sequences are collected for many other gene families (from novel sequencing projects etc). Is it meant (only) in human genome? If so, authors shall ultimately give ALL GenBank accession numbers (or UniProt accessions) in a Table to get a clear overview for the reader.

Line 90: The term "peculiar retention" shall be clarified (in evolutionary content?)

Results and discussion lines 95-96: As already mentioned for the abstract, better "similarities in structural fold" shall be used here.

Lines 101-102: "103 sequences from across the animal kingdom and IL-1 family members" - this is quite confusing. In line 359 on page 12_107 sequences are mentioned. Why is there such discrepancy? Were 4 sequences mentioned in line 359 omitted from the final analysis? Moreover, it needs to be clearly defined how many of these 103 (or 107?) sequences are indeed IL-1 family members? For a typical evolutionary analysis paper ALL such sequences need to be annotated in a Table with their accession codes from public databases. Nowadays, it is usual to include much more sequences than 103 in a comprehensive phylogenetic analysis so the authors also need to indicate if those sequences that they have selected really comprehensive in the term of (possibly) equal distribution among all known mammalian orders.

Lines 105-106: The (proposed) common ancestor shall be indicated in Figure 1B. Is it a sequence motif corresponding with *Callorhinchus milii*? There are, however, some differences in the architecture if compared with mammals. Currently existing sequence from a Cartilaginous fish cannot be a real common ancestor (if strictly speaking & writing) but only can probably have the nearest evolutionary distance to a proposed last common ancestor of this entire clade that is at the organism level in our times probably extinct.

Lines 120-121: "a more accurate IL-1 family tree" Was such a tree rebuilt from a preliminary tree by optimising some parameters of the phylogenetic method? Is this that one presented in Figure 1?

Line 129: "independent evolutionary events are not unlikely" This shall be re-written in a more understandable way, citing also a newer literature on convergent type of evolution (independent evolutionary events leading to very similar properties).

Lines 133-134: In Figure 1 "analyses of IL-1 family" is mentioned but in Supplementary Figure 1

the analysis of the "IL-1 superfamily clusters" is presented. This is somehow confusing. Please clarify the inclusion criteria into IL-1 family and what shall be understood under "IL-1 superfamily clusters"?

Line 137: what is the measure of "approximately equally unrelated to"... can it be quantified by evolutionary distances obtained from the presented ML tree? However, the bootstrap values in many nodes of Figure 1A (and also in Suppl. Fig.1) are rather low. As a generally accepted rule mentioned also by authors, only bootstrap values (or percentage of trees...) ABOVE 50 shall be presented as seriously supported. Probably the most interrogative is the value 0 located at the outer clade in Figure 1A. Does this stand for 0% of trees with such association of analysed groups?

Line 272: It shall be specified for the reader in which part of Figure 1a (or Supplementary Figure 1?) the toothed whale clade can be found. Figure 3 where it is shown is just a schematic presentation. Moreover, for most readers it is not obvious in which extent the NLS is lacking in comparison to other clades? Can it be clearly presented in some kind of multiple sequence alignment?

Lines 280-282: Similar as in line 272, insight in such a sequence comparison is needed.

Line 293: Can the selective force of divergence between IL-1 α & IL-1 β be demonstrated or commented also with respect to the intermediate clade IL-33 seen in Figure 1A?

Line 294: "HAX-1 is an ancient protein" This is from the phylogenetic point of view not a good expression. HAX-1 may have ancient roots but it is occurring also in modern animals. The reconstruction of its ancestral sequence would be possible from contemporary sequences and their phylogenetic relationships.

Lines 307-308: Also the effort on phylogenetic analysis should be "refocused" as there are again low bootstrap values (percentages) in some nodes of Suppl. Fig.2.

Conclusion line 327: "Central and founding member is IL-1b" – as already mentioned for Abstract line 26, this formulation is unusual for phylogenetic analysis and shall be better redefined as "monophyletic clade" if it is well supported from the phylogenetic tree (in this version does not seem to as there are really low percentages on some clades).

Line 329: "do not share a common ancestor" – if the reconstructed tree presented in Figure 1a is statistically not well supported this also could be questionable. It just can be presented as a working hypothesis that needs more support.

Methods lines 352-353: which are the "established IL-1 family sequences"? As mentioned above an overview Table with all used sequences with labelling of the "established ones" is really needed.

Line 353: "The evolutionary history was inferred by..." at this point it is necessary to specify whether genomic DNA, cDNA or protein sequences were used with the explanation why the particular type of sequences was preferred for the phylogenetic analysis.

Line 359: 107 or 103 sequences? (see also above)

Line 360: Today the authors can use also a new version namely MEGA 7 that has some advantages against MEGA 6. This consideration is left for author's decision but anyway, it is absolutely necessary to prove whether the best substitution matrix was used in order to get the best possible statistical support to justify the hypothesis that the authors want to spread. Alternatively, it is recommended that authors use also an alternative molecular evolutionary analysis package like Mr. Bayes and compare the output with the (hopefully improved) MEGA output.

Line 375: Authors shall give the name of the software suite (and the corresponding company) that was used for the computation of synonymous/nonsynonymous substitution statistics.

Figure 1, line 558: were there 103 or 107 amino acid sequences used for phylogeny? (mentioned also above). Why only 59 positions were used in the final dataset?

Figure 2, line 564: how is a "conserved amino acid residue" defined? Multiple sequence alignment shall help to illustrate this. Or is this the same conservation given in Figure 3? Needs to be clarified.

Supplementary Figure 1 what is the difference to Figure 1A? Is suppl. fig.1 just a detail of Figure 1A or was there a completely new analysis performed (IL-1 family versus IL-1 superfamily clusters).

Reviewer #3 (Remarks to the Author):

This study explores the evolutionary origin of the IL-1 family members. The study arrives at the conclusion that they did not originate from gene duplication. Consequently, the authors suggest that IL-18 and IL-33 should no longer be considered part of the IL-1 family. While the understanding of their evolutionary origin is academically interesting, this reviewer is not sure why this is biologically significant and how this would impact how we are studying these proteins. Since the IL-1 family "issue" is not specific to the cytokines but also involves the receptors and accessory proteins, impact could be improved by including analyses of how this protein family co-evolved with the cytokines. Another interesting angle could be to extend the link to the FGF family.

In many places, it is unclear what has been done by the authors and what has been done by others. The study would greatly benefit from separation of the Results and Discussion sections.

References 1-3: References to the joint efforts of the scientific community in naming the "IL-1 family members" should be included. IL-18 and IL-33 were included in the family for many reasons. Why is it important from a biological point of view (not conceptual evolutionary understanding) to say they are not part of the family?

References to studies that originally proposed the idea that the IL-1 family members arose through gene duplication should be included. References 7-9 are inappropriate in this context.

Reference 10-11 are inappropriate in the context used. They are from 1992 and 1996, which is before discovery of many of the "IL-1 family members".

References need to be added in first paragraph, page 2 of Introduction.

Results section: Some of the evolutionary analysis appears to have been done before – References 20, 21, 23 and 24 are provided.

Second paragraph, Results section: The jump from "why does IL-1 α , a structurally and functionally similar protein to IL-1 β that formed from a gene duplication event of IL-1 β , have such a dissimilar sequence?" to talking about the antagonist is confusing. I don't follow the argument.

Why reference 26 (title "Systematic identification of cell cycle-dependent yeast nucleocytoplasmic shuttling proteins by prediction of composite motifs.") is included in the context it is used is unclear.

It is stated "there is a group of animals that contains the highly conserved pro-domain of IL-1 α yet lack an NLS" however, no data is provided. Figures should be added to show which animals these are. Or is that actually the NEXT paragraph? If that is the case the manuscript should be rewritten to present the data in a more linear fashion. Showing how the NLS is "absent" in certain species would be helpful. Again, separating Results from Discussion would help making it easier to read, understand, and evaluate novelty of the study.

Minor issues:

Abstract line 7: IL-1 α , IL-18 and IL-33 are discrete single proteins, not subfamilies.

Use of a, b and c versus alpha, beta and gamma is inconsistent.

Reviewers' comments

Reviewer #1 (Remarks to the Author):

Major comments:

-Concerning IL-18 and IL-33, the authors propose that structural similarity is not sufficient to include them in the IL-1 family, and indeed they show that sequence homology with the other members is low. Thus evolutionary evidence would suggest the exclusion of IL-33 and IL-18 from the IL-1 family. This analysis is important for better understanding the evolution and biology of these cytokines. However, the authors should take into consideration functional and biological homologies that justify or explain their inclusion in the IL-1 family (e.g. the homology among the receptors of IL-1 family, including IL-18R α and IL-18R β , or ST2 which couples with IL-1RACP to signal; consider that IL-37 signals through IL-18R α , that all the members of the 4 clusters share a common mode of activation through cleavage of the pro domain and activate a shared signalling pathway).

Author response:

We would like to thank the reviewer for highlighting the lack of discussion of the definition of family with regards to proteins and genes, as well as, the need to discuss the ligands in the context of the receptors.

We have now performed a very thorough analysis of the receptor evolution (new Figure 1). This has involved 231 sequences from across the animal kingdom and 10 receptor homologues. We have also performed chromosomal anatomy analysis for the most in-depth investigation into IL-1 receptor evolution to-date. Performing this has provided a greater understanding of both the receptor and ligand evolution. As an example, the lack of sequence difference between mammalian and non-mammalian IL1R2 supports the hypothesis that the driving force of IL-1 α evolution is not differences in IL1R2 binding. As IL-1 α is purely mammalian a co-evolution acceleration would be expected and a greater difference between mammalian IL1R2 and non-mammalian IL1R2 would be expected.

We have also included additional paragraphs that put a greater emphasis on how the inclusion or exclusion of IL-18 and IL-33 largely depends on whether the original definition of gene/protein family (Britten and Kohne 1968) was used which is based on common ancestry, or, the more loosely defined definition of family is used based on commonality of function, sequence and protein interactions. We have highlighted the importance of this difference in definition and conflating the two has very important biological implications. For example, understanding that there is little evidence of common ancestry between IL-33 and IL-18 ligands, and the rest of the IL-1 family is critical when considering modelling the ligand- receptor interactions. Two papers have been released on IL-18 and IL-33 receptor interactions and in both of these the modelling used the interactions of IL-1 β and IL-1R1 as a modelling scaffold. This validity of this model depends on the conserved nature of this interaction assuming common ancestry of the receptors and ligands. Our finding suggest that this approach may be valid because of the close conserved nature of the receptors, however, we also suggest that because the ligands likely do not share a recent common ancestor there is now greater doubt of these publish models accuracy.

-The authors propose that HAX-1 binding is the major directional selective force pushing the divergence of IL-1 α from IL-1 β , based on the high conservation of the regions which bind HAX-1 in IL-1 α pro domain. This hypothesis is novel and interesting. However the authors do not show functional data supporting it. The authors should comment on the relationship between HAX-1 binding and the described differences in function of IL-1 β and IL-1 α .

Author response:

We have included a more in-depth discussion of the HAX-1 IL-1 α binding literature highlighting the potential inflammatory effect of the HAX-1 interaction as demonstrated by Kawagushi et al. 2004 using siRNA knockdown. We have also added additional biological data showing that there is a specific interaction between pro-IL-1 α and HAX-1 which doesn't occur with mature IL-1 α or IL-1 β (new Figure 4D) fully supporting our evolutionary predicted hypothesis. We are continuing to follow this line of investigation and are submitting grant applications which will allow us to publish a full investigation of HAX-1 and IL-1 α interactions.

HAX-1 is introduced in the manuscript abruptly (line 292) not showing results, but discussing its potential relevance. Lines 301-303 should be presented first.

Author response:

HAX-1 is now appropriately introduced as suggested. We have also now included it in the introduction of the manuscript.

-They also propose that gene regulation through histone acetyltransferase complex binding may have been involved in pushing the divergence of IL-1 α from IL-1 β . It is not clear for the reader whether this hypothesis is considered solid to the authors.

Author response:

We have now clarified this in the text to suggest that we consider it less likely to have been the major driving force due to the lack of a NLS in the IL-1 α sequences of toothed whales yet the HCPR is still conserved which contains the HAX-1 binding domain. We propose that this suggests the major evolutionary pressure will have come outside the nucleus.

Minor comments

- HCLS1 should be spelled out.

Author response:

HCLS1 and HAX-1 are now fully defined in the text.

- Not all figure panels are mentioned in the text.

Author response:

This has now been addressed.

Reviewer #2 (Remarks to the Author):

Points to consider & improve:

Abstract, line 24: In content of an evolutionary publication it is rather vague to use the term "sequence homology". Either it is just only a sequence similarity or it is really an evolutionary homology. The chaotic usage of such terms was criticised previously many times by many reviewers in evaluation of papers aimed to describe evolutionary relationships. (Please consider this aspect also in the whole manuscript).

Author response:

Respectfully, 'sequence homology' refers to sequences being evolutionary related without any information about how related they are, while 'sequence similarity' is used to infer evolutionary

relatedness, i.e., homology. The referee is referring to confusion when authors have incorrectly talked about amounts of sequence homology, whereas it is only correct to refer an amount of sequence similarity and from this infer sequence homology. We have been careful not to confuse these concepts.

Abstract, line 26: The term “founding member” is really strange for a phylogenetic analysis. Better “basal clade” (if it really is one) or “monophyletic group” (if more clades are phylogenetically joined together leading to a common ancestor) shall be used - also throughout the whole manuscript. Abstract, line 29: “no evidence of common ancestry” – brief details for this shall be given in the abstract

Author response:

We agree ‘founding member’ can be confusing, and has been removed from the manuscript. Respectfully, however, ‘basal clade’ is not an appropriate term also as it’s open to misinterpretation, discussed here <<http://onlinelibrary.wiley.com/doi/10.1111/j.0307-6970.2004.00262.x/full>> and here <<http://for-the-love-of-trees.blogspot.co.uk/2016/09/the-ancestors-are-not-among-us.html>>, and monophyletic group does not have the meaning we require for an ancestral gene. We now use the term ‘shared common ancestor’ to be clear.

Introduction line 39: “There are 11 members of the IL-1 family”... so only eleven is really a very few in times where hundreds of sequences are collected for many other gene families (from novel sequencing projects etc). Is it meant (only) in human genome? If so, authors shall ultimately give ALL GenBank accession numbers (or UniProt accessions) in a Table to get a clear overview for the reader. Line 90: The term “peculiar retention” shall be clarified (in evolutionary content?)

Author response:

We have adjusted our terminology to clarify that we protein/gene family members, not species homologues. We have also constructed tables which of all the genes with GeneIDs and sources and included this as supplementary table 1. We have expanded on the term “peculiar retention”, to explain that this is peculiar because the ligands are very similar. They are similar in size, expression, activation pathway and activity. The retention of a seemingly redundant protein in all mammalian species is unusual unless there is a unique role for the protein.

Results and discussion lines 95-96: As already mentioned for the abstract, better “similarities in structural fold” shall be used here.

Author response:

This has been changed to similar protein fold as suggested.

Lines 101-102: “103 sequences from across the animal kingdom and IL-1 family members” – this is quite confusing. In line 359 on page 12_107 sequences are mentioned. Why is there such discrepancy? Were 4 sequences mentioned in line 359 omitted from the final analysis? Moreover, it needs to be clearly defined how many of these 103 (or 107?) sequences are indeed IL-1 family members? For a typical evolutionary analysis paper ALL such sequences need to be annotated in a Table with their accession codes from public databases. Nowadays, it is usual to include much more sequences than 103 in a comprehensive phylogenetic analysis so the authors also need to indicate if those sequences that they have selected really comprehensive in the term of (possibly) equal distribution among all known mammalian orders.

Author response:

This appears to have been a typographical error, however, to improve the confidence of our tree we have now included more sequences and rerun the alignment using MEGA 7. From this we now have 155 amino acid sequences. This has been amended throughout the manuscript.

Lines 105-106: The (proposed) common ancestor shall be indicated in Figure 1B. Is it a sequence motif corresponding with *Callorhinchus milii*? There are, however, some differences in the architecture if compared with mammals. Currently existing sequence from a Cartilaginous fish cannot be a real common ancestor (if strictly speaking & writing) but only can probably have the nearest evolutionary distance to a proposed last common ancestor of this entire clade that is at the organism level in our times probably extinct.

Author response: As IL-1β homologues are present in all vertebrate clades it is clear that a proto-IL-1β was likely present in the common ancestor of all vertebrate. It is not accurate to say that Callorhinchus has the common ancestor sequence as it is likely to be as different to the proto-IL-1β as the other clades as the Callorhinchus clade has had the same period of time to drift in sequence homology from the proto-IL-1β. We have addressed this in the manuscript.

Lines 120-121: “a more accurate IL-1 family tree” Was such a tree rebuilt from a preliminary tree by optimising some parameters of the phylogenetic method? Is this that one presented in Figure 1?

Author response:

This refers to Fig 1D (now Fig 2D). This tree was constructed based on a combination of the sequence similarity tree in Fig 1A (now Fig 2A) as well as the chromosomal anatomy and knowledge of clade evolution. By combining these sources of information the Fig 1D (now Fig 2D) tree was constructed. However, we acknowledge that this language is perhaps a bit strong and so we have changed it to “a more probably IL-1 ligand superfamily tree that better predicts the evolutionary history of protein/gene families”.

Line 129: “independent evolutionary events are not unlikely” This shall be re-written in a more understandable way, citing also a newer literature on convergent type of evolution (independent evolutionary events leading to very similar properties).

Author response:

This has been changed to “supporting the plausibility of independent evolutionary events”. New citations of convergent protein evolution have been included. Specifically the Nature review paper by Storz 2016 which posits the likelihood of convergent evolution is inversely proportional to the complexity and specificity of the protein structure and function.

Lines 133-134: In Figure 1 “analyses of IL-1 family” is mentioned but in Supplementary Figure 1 the analysis of the “IL-1 superfamily clusters” is presented. This is somehow confusing. Please clarify the inclusion criteria into IL-1 family and what shall be understood under “IL-1 superfamily clusters”?

Author response:

Thanks you for picking up this inconsistency. We have now included specific definition for family and superfamily in the manuscript and have adhered to these definitions throughout.

Line 137: what is the measure of “approximately equally unrelated to”... can it be quantified by evolutionary distances obtained from the presented ML tree? However, the bootstrap values in many nodes of Figure 1A (and also in Suppl. Fig.1) are rather low. As a generally accepted rule mentioned also by authors, only bootstrap values (or percentage of trees...) ABOVE 50 shall be

presented as seriously supported. Probably the most interrogative is the value 0 located at the outer clade in Figure 1A. Does this stand for 0% of trees with such association of analysed groups?

Author response:

Essentially this is a key point in the manuscript. Construction of this tree and using the tree alone as evidence is not advisable because it includes proteins which do not share a common ancestor and the short and divergent nature of the data leave a relatively weak phylogenetic signal. Hence the analysis is struggling to find consistent clustering in trees during the bootstrapping process. That is why we used chromosomal anatomy and integrated this with sequence similarity and knowledge of clade evolution to construct the summary tree in Figure 1D. We have outlined this composite process more in the methods of the manuscript. We have also included more sequences and rerun the analysis on MEGA 7 to improve our confidence in the divergent patterns of related proteins. We have also repeated the tree analysis after breaking the tree up into our proposed families of related proteins and included these in supplementary Figures 3-6. From this it can be seen that the number of positions used in the analysis and confidence of the branching is higher.

Line 272: It shall be specified for the reader in which part of Figure 1a (or Supplementary Figure 1?) the toothed whale clade can be found. Figure 3 where it is shown is just a schematic presentation. Moreover, for most readers it is not obvious in which extent the NLS is lacking in comparison to other clades? Can it be clearly presented in some kind of multiple sequence alignment?

Author response:

Thank you for this suggestion. We have now included an IL-1a tree in the supplement with the toothed whales highlighted (supplementary Figure 3), you will see no dramatic acceleration in sequence divergence despite the lack of NLS. We have also included in a sequence alignment in new Figure 4B highlighting the lack of NLS and what key amino acid changes have occurred in the toothed whale.

Lines 280-282: Similar as in line 272, insight in such a sequence comparison is needed.

Author response:

This has been illustrated and expanded on, as described above.

Line 293: Can the selective force of divergence between IL-1 α & IL-1 β be demonstrated or commented also with respect to the intermediate clade IL-33 seen in Figure 1A?

Author response:

We have now commented on the potential for IL-1a sharing a common ancestor with IL33 or IL18. However, we propose this unlikely due to the chromosomal anatomy evidence and the existence of IL-18 across all clades, while IL-1a is present in mammals.

Line 294: "HAX-1 is an ancient protein" This is from the phylogenetic point of view not a good expression. HAX-1 may have ancient roots but it is occurring also in modern animals. The reconstruction of its ancestral sequence would be possible from contemporary sequences and their phylogenetic relationships.

Author response:

We have reworded this sentence to describe the period of likely first appearance.

Lines 307-308: Also the effort on phylogenetic analysis should be "refocused" as there are again low bootstrap values (percentages) in some nodes of Suppl. Fig.2.

Author response:

There is only one hemichordate HAX-1 sequence. This has prevented this branch from finding a stable position in the tree. The point of this tree is not to posit the potential evolutionary history, but to establish that all clades have a HAX-1 homologue, suggesting it first appeared prior to the formation of the vertebrate clade.

Conclusion line 327: “Central and founding member is IL-1b” – as already mentioned for Abstract line 26, this formulation is unusual for phylogenetic analysis and shall be better redefined as “monophyletic clade” if it is well supported from the phylogenetic tree (in this version does not seem to as there are really low percentages on some clades).

Author response:

Thank you for rightly picking up this loose terminology. We have addressed this.

Line 329: “do not share a common ancestor” – if the reconstructed tree presented in Figure 1a is statistically not well supported this also could be questionable. It just can be presented as a working hypothesis that needs more support.

Author response:

As discussed above, we have expanded on this to explicitly say that the Fig. 1A (now Fig 2A) is not likely due to the inclusion of non-family members (IL-18 and IL-33). Hence the construction of Fig.1D (now Fig 2D) which uses chromosomal evidence and clade evolution to construct a more likely tree.

Methods lines 352-353: which are the “established IL-1 family sequences”? As mentioned above an overview Table with all used sequences with labelling of the “established ones” is really needed.

Author response:

This table has now been included as supplementary table 1. Thank you for the suggestion.

Line 353: “The evolutionary history was inferred by...” at this point it is necessary to specify whether genomic DNA, cDNA or protein sequences were used with the explanation why the particular type of sequences was preferred for the phylogenetic analysis.

Author response:

We have now placed extra emphasis on when protein or genomic evidence was used for each analysis.

Line 359: 107 or 103 sequences? (see also above)

Author response:

Thank you for picking this up. It is now 155 sequences and has been updated throughout the manuscript.

Line 360: Today the authors can use also a new version namely MEGA 7 that has some advantages against MEGA 6. This consideration is left for author’s decision but anyway, it is absolutely necessary to prove whether the best substitution matrix was used in order to get the best possible statistical support to justify the hypothesis that the authors want to spread. Alternatively, it is recommended that authors use also an alternative molecular evolutionary analysis package like Mr. Bayes and compare the output with the (hopefully improved) MEGA output.

Author response:

We have now reanalysed all of the data using MEGA7 as suggested.

Line 375: Authors shall give the name of the software suite (and the corresponding company) that was used for the computation of synonymous/nonsynonymous substitution statistics.

Author response:

This is now MEGA7 and this detail has now been added to the manuscript.

Figure 1, line 558: were there 103 or 107 amino acid sequences used for phylogeny? (mentioned also above). Why only 59 positions were used in the final dataset?

Author response:

This was a typo. We have now updated it to the 155 sequences used in the new analysis. In the new analysis 64 positions were used. The smaller family members such as IL-1RN have ~150 a.a. in their sequence. There are obviously deletions and additions for different proteins resulting in a smaller proportion of the protein used for the sequence alignment. This is done during the JTT modelling by MEGA7.

Figure 2, line 564: how is a “conserved amino acid residue” defined? Multiple sequence alignment shall help to illustrate this. Or is this the same conservation given in Figure 3? Needs to be clarified.

Author response:

This is described in the methods. It is the percentage of species with the modal amino acid at each aligned site.

Supplementary Figure 1 what is the difference to Figure 1A? Is suppl. fig.1 just a detail of Figure 1A or was there a completely new analysis performed (IL-1 family versus IL-1 superfamily clusters).

Author response:

This was a completely new analysis. We have included more sequences now and established that FGF, IL8, IL18 and IL33 are similarly unrelated to IL1b. Hence, this supports the exclusion of IL18 and IL33 from the ancestral family.

Reviewer #3 (Remarks to the Author):

This study explores the evolutionary origin of the IL-1 family members. The study arrives at the conclusion that they did not originate from gene duplication. Consequently, the authors suggest that IL-18 and IL-33 should no longer be considered part of the IL-1 family. While the understanding of their evolutionary origin is academically interesting, this reviewer is not sure why this is biologically significant and how this would impact how we are studying these proteins. Since the IL-1 family “issue” is not specific to the cytokines but also involves the receptors and accessory proteins, impact could be improved by including analyses of how this protein family co-evolved with the cytokines.

Author response:

Thank you for the suggestions. We have now included several points to outline the biological significance of understanding evolutionary history of proteins. There are three main points: (i) Understanding protein/gene ancestry is crucial for biologically relevant investigations such as ligand protein interaction modelling as discussed above. Indeed both IL-18 and IL-33 receptor interactions were modelled using IL1R1 and IL-1b interaction a search model. Meaning IL1R1 structure was used

as a scaffold in modelling the IL18R1 and IL1RL1 ligand interaction. This approach is most valid in the context of conserved structure in evolutionarily related proteins. (ii) As explored with IL-1a in this research, conservation of subdomains help delineate which physiological functions are most likely to be of physiological importance and which maybe unimportant spurious interactions. For example if in vitro protein A binds to proteins B and C, one can delineate which interaction is likely of physiological importance based on the conservation of those binding domains. (iii) Understanding evolutionary history is of scientific interest in itself.

As per your suggestion we have also performed extensive analysis on the receptor and binding proteins constructing the most in-depth evolutionary tree for publish utilizing 231 sequences. This was an excellent suggestion as it supported our conclusion on IL-1a subfunctionalisation. As mammalian IL1R2 did not show accelerated divergence in the mammal species this implies that the differential binding of IL1R2 to IL-1b and IL1a did not drive the divergence of IL-1a, thus supporting the HAX-1 binding hypothesis.

Another interesting angle could be to extend the link to the FGF family.

Author response:

We would like to thank the reviewer for this suggesting as a more thorough investigation into the FGF family provided significant evidence for the proposed separation of the IL-1 ancestral family. We have expanded the FGF tree substantially and it is presented in supplementary Figure 1. We included in this tree all FGF molecules that have had a crystal structure produce and were found to have the beta trefoil structure. This involved constructing a tree of 344 sequences of 7 FGF molecules and the IL-1 superfamily members. This tree clearly shows that IL-18 and IL-33 are equally related (or unrelated) to the FGF family as the IL-1 family. The maximum likelihood tree even placed IL-33 and IL-18 on FGF family side of the tree, however, the tree is highly unstable. This instability is expected when constructing a tree of unrelated proteins. This provides substantial evidence for the splitting off of IL-18 and IL-33 from the IL-1 ancestral family.

In many places, it is unclear what has been done by the authors and what has been done by others.

Author response:

Thank you for the suggestion, it is very important that work is properly attributed. We have separated the discussion from the results and addressed any ambiguity in the text. We believe original research is now clearly delineated.

The study would greatly benefit from separation of the Results and Discussion sections.

Author response:

We agree and have now separated the results and discussion sections.

References 1-3: References to the joint efforts of the scientific community in naming the “IL-1 family members” should be included. IL-18 and IL-33 were included in the family for many reasons. Why is it important from a biological point of view (not conceptual evolutionary understanding) to say they are not part of the family?

Author response:

We have now included in-depth discussion on ancestral vs functional family. We have distinguished between the two using the terms ancestral family and superfamily. As discussed above we have also included clear examples where understanding ancestral relationship is critical to how we investigate the proteins today (such as the protein modelling example). We have also included the more citations

for work that has contributed to establishing nomenclature of IL-1 as well as including key examples where high impact literature has conflated functional families with ancestral families. This demonstrates the need for this research to clearly outline the evidence for ancestral relationships between IL1 ligand and receptor family members.

References to studies that originally proposed the idea that the IL-1 family members arose through gene duplication should be included. References 7-9 are inappropriate in this context.

Author response:

These references have been updated.

Reference 10-11 are inappropriate in the context used. They are from 1992 and 1996, which is before discovery of many of the “IL-1 family members”.

Author response:

These were included as they established the structure of the proteins which were then used to group IL18 and IL33 in the IL1 family. However, we should have included the papers which defined the family members. This has now been done. Thank you for the suggestions.

References need to be added in first paragraph, page 2 of Introduction.

Author response:

Appropriate references have now been added.

Results section: Some of the evolutionary analysis appears to have been done before – References 20, 21, 23 and 24 are provided.

Author response:

Some analyses have been performed previously with few numbers of animals, but not to the extent or scale presented here and because of these limitations key implications were not described such as the independent evolution of IL-33 and IL-18, and the divergent evolution of IL-1 α . This research has important implications on previously published research such as 3D models constructed assuming close evolutionary relationships between ligands and receptors. It also highlights new avenues of research predicting non-IL-1R1 functions of IL-1 α which we highlight. This thus represents the most comprehensive analysis of the IL-1 family to date, redefining the family, providing a framework for future research in the field.

Second paragraph, Results section: The jump from “why does IL-1 α , a structurally and functionally similar protein to IL-1 β that formed from a gene duplication event of IL-1 β , have such a dissimilar sequence?” to talking about the antagonist is confusing. I don’t follow the argument.

Author response:

We have now expanded on this. This was to highlight that if IL-1 α evolved to be constitutively active it could have undergone a truncation mutation similar to IL1RN. As this has not happened, the constitutive activity of IL-1 α is unlikely the cause of the divergence.

Why reference 26 (title “Systematic identification of cell cycle-dependent yeast nucleocytoplasmic shuttling proteins by prediction of composite motifs.”) is included in the context it is used is unclear.

Author response:

This reference is used for the NLS scores. These scores are highly predictive of nuclear localization across a range of clades (from yeast to mammalian cells), demonstrating the highly conserved nature of this process. However, we see that this reference has been placed incorrectly for the two sentences you point out. We have replaced these with the appropriate references.

It is stated “there is a group of animals that contains the highly conserved pro-domain of IL-1 α yet lack an NLS” however, no data is provided. Figures should be added to show which animals these are. Or is that actually the NEXT paragraph? If that is the case the manuscript should be rewritten to present the data in a more linear fashion. Showing how the NLS is “absent” in certain species would be helpful.

Author response:

Thank you for this suggestion. We have rewritten this paragraph to introduce the toothed whale clade as lacking the NLS. We have also included in figure 4 a sequence alignment of the toothed whale clade to highlight which amino acids have changed in the NLS and how this has affected their likelihood of NLS translocation through the NLS signal peptide.

Again, separating Results from Discussion would help making it easier to read, understand, and evaluate novelty of the study.

Author response:

We agree and have now separated the results and discussion sections.

Minor issues:

Abstract line 7: IL-1a, IL-18 and IL-33 are discrete single proteins, not subfamilies.

Author response:

This has now been corrected and subfamilies has been changed to proteins.

Use of a, b and c versus alpha, beta and gamma is inconsistent.

Author response:

We have now formatted all the images and text to keep this consistent.

Reviewer #1 (Remarks to the Author):

The authors addressed the points I raised

Reviewer #2 (Remarks to the Author):

Revised publication:

Redefining the interleukin 1 family from an evolutionary perspective
by Rivers-Auty, Daniels, Colliver, Robertson and Brough.

Authors have completely revised their publication on IL-1 superfamily evolution and removed most shortcomings. However, some minor points still need to be considered.

Suggestion: minor revision

Points to consider & improve:

If, after an intensive work the authors come to the (important) conclusion that "the term superfamily better describes our current understanding of the evolutionary history of the IL-1 superfamily members" then they should accordingly CHANGE THE TITLE of their publication – and this is an important point after its acceptance for all search engines in the web. It will also be helpful if authors draw an overview scheme (somewhere at the beginning of their article) on the previously described members of this newly constituted superfamily.

Introduction lines 51-53: The issue on loosened definition of gene & protein families and sharing of a common ancestry is very interesting also beyond this particular publication but in the revised introduction it is shored with a list of rather old publications: #6 from 1997; #7 even from 1968; #8 from 2005. Can the authors make an update by presenting (also) most recent publications to this topic? Sure, from the phylogenetic point of view a gene family only makes sense if there is an evidence for a common ancestor... For the superfamily definition there may be a very ancient common ancestor and some missing links between extant members or a convergent type of evolution. Anyway, there must be some exact criteria (common functional aspects for a superfamily) that can be mentioned here briefly.

Results

Page 6 and Figures 1A, 2A – the output of MEGA7 phylogeny is presented in a very small scale in figures 1,2. E.g. to inspect the obtained bootstrap values one has to use magnifying glass or an extreme zoom. Can the authors make a different presentation of these important trees e.g. in a circular form on a whole A4 page with appropriate font size? The supplementary figures are OK in this respect.

Pages 6-7 the time estimation of the evolutionary events – authors quote again a rather old publication of Kumar & Hedges from 1998 based on 658 genes from 207 vertebrate species. However, there is an update from the same authors Hedges S.B. et al. Mol. Biol. Evol. 32(4):835-845 in 2015 based on 50,632 species that could be considered for a possible update of this part.

Methods lines 513-514 – the maximum likelihood method based on the JTT matrix model was used for important conclusions in the text. Was this indeed the best substitution model for used sequences as can be evaluated in the MEGA package? Or at least was it among the best ones if not the very best? Authors shall comment on this (present a supplementary material on this issue with overview on potential matrices). The statement that "the phylogenetic inference of these trees was intrinsically unreliable" – see line 521 is interesting and indeed, in the supplementary figures those trees that are based on longer alignments (more positions considered) have significantly higher bootstrap values. But when we now come back to the discussion - like that in lines 338-339 that

IL-33 and IL-18 do not share a common ancestor – so are such conclusions based on very low “percentage of trees clustering in this way”? (i.e. bootstraps cf. lines 848-849 for explanation). So this is something like a negative evidence. It would be interesting to consider whether there are some missing common ancestors (of the whole IL-1 superfamily) in some basal evolutionary clade that evolved before the segregation of vertebrates? Authors can comment on this shortly.

Line 925 and Figure 4B – “cNLS scores” shall be explained in the figure legend appropriately, not just as a citation of a reference.

Responses to the reviewer

Reviewer #2 (Remarks to the Author):

If, after an intensive work the authors come to the (important) conclusion that “the term superfamily better describes our current understanding of the evolutionary history of the IL-1 superfamily members” then they should accordingly CHANGE THE TITLE of their publication – and this is an important point after its acceptance for all search engines in the web.

Author response: The title has now been changed accordingly.

It will also be helpful if authors draw an overview scheme (somewhere at the beginning of their article) on the previously described members of this newly constituted superfamily.

Author response: Fig 2 has now been revised to include an overview scheme of the ancestral and superfamily members of the ligands.

Introduction lines 51-53: The issue on loosened definition of gene & protein families and sharing of a common ancestry is very interesting also beyond this particular publication but in the revised introduction it is shored with a list of rather old publications: #6 from 1997; #7 even from 1968; #8 from 2005. Can the authors make an update by presenting (also) most recent publications to this topic? Sure, from the phylogenetic point of view a gene family only makes sense if there is an evidence for a common ancestor... For the superfamily definition there may be a very ancient common ancestor and some missing links between extant members or a convergent type of evolution. Anyway, there must be some exact criteria (common functional aspects for a superfamily) that can be mentioned here briefly.

Author response: The age of these citations represent the first groups to outline the definitions of gene and protein families. We have now included the most up-to-date review on the general subject of superfamilies (Han 2007). Unfortunately, this review comes to the same conclusion as our manuscript that the terms “family” and “superfamily” are ill defined with several databases using these terms in completely different fashions. This is why we explicitly state how these terms will be used in our paper (lines 56-57 and 85-87).

Page 6 and Figures 1A, 2A – the output of MEGA7 phylogeny is presented in a very small scale in figures 1,2. E.g. to inspect the obtained bootstrap values one has to use magnifying glass or an extreme zoom. Can the authors make a different presentation of these important trees e.g. in a circular form on a whole A4 page with appropriate font size? The supplementary figures are OK in this respect.

Author response: Figs 1A and 2A have now been removed from Figs 1 and 2 and are available as enlarged legible versions as Supplementary Figs 1 and 2. The text has been modified as appropriate to accommodate this change.

Pages 6-7 the time estimation of the evolutionary events – authors quote again a rather old publication of Kumar & Hedges from 1998 based on 658 genes from 207 vertebrate species. However, there is an update from the same authors Hedges S.B. et al. Mol. Biol. Evol. 32(4):835-845 in 2015 based on 50,632 species that could be considered for a possible update of this part.

Author response: We have now included the updated reference.

Methods lines 513-514 – the maximum likelihood method based on the JTT matrix model was used for important conclusions in the text. Was this indeed the best substitution model for used sequences as can be evaluated in the MEGA package? Or at least was it among the best ones if not the very best? Authors shall comment on this (present a supplementary material on this issue with overview on potential matrices). The statement that “the phylogenetic inference of these trees was intrinsically unreliable” – see line 521 is interesting and indeed, in the supplementary figures those trees that are based on longer alignments (more positions considered) have significantly higher bootstrap values. But when we now come back to the discussion - like that in lines 338-339 that IL-33 and IL-18 do not share a common ancestor – so are such conclusions based on very low “percentage of trees clustering in this way”? (i.e. bootstraps cf. lines 848-849 for explanation). So this is something like a negative evidence. It would be interesting to consider whether there are some missing common ancestors (of the whole IL-1 superfamily) in some basal evolutionary clade that evolved before the segregation of vertebrates? Authors can comment on this shortly.

Author response: We reran the trees using the more recently developed Whelan and Goldman, and Dayhoff matrix based models and found only subtle changes in the log likelihood values and no meaningful changes in the structure of the trees. We have included this extra step in the methods.

We have shown that the instability of the full ligand tree is due to the inclusion of IL18 and IL33. We gain greater stability and include more amino acid positions when IL18 and IL33 are analysed in isolation. We consider this evidence against their inclusion in the family. We supported this by showing that IL18 and IL33 are equally (if not more) related to proteins not in the ancestral family of IL1 (specifically FGF proteins). Furthermore, we showed that there is no chromosomal anatomy evidence supporting their common ancestry, despite the relatively recent appearance of IL33 on the evolutionary timescale (as it is purely a mammalian protein).

The hypothesis that a proto-IL-1 protein is the common ancestor of all IL-1 superfamily members and that the apparent outliers of IL18 and IL33 may be due to extinct lineages that originally evolved from a common proto-IL-1 protein is interesting. However, it is speculative. For example it proposes that an IL33 lineage protein was present in all vertebrates but was lost without a trace in the cartilaginous fishes, bony fishes, birds and reptiles, but not the mammals. We believe the unrelated protein hypothesis has more evidence to support it and is more parsimonious.

Line 925 and Figure 4B – “cNLS scores” shall be explained in the figure legend appropriately, not just as a citation of a reference.

Author response: This has now been amended as suggested.